# Sensitivity of ID NOW and RT–PCR for detection of SARS-CoV-2 in an ambulatory population

Yuan-Po Tu[1], Jameel Iqbal[2]*, Timothy O'Leary[3,4]

[1]Urgent Care, The Everett Clinic-Part of Optum, Everett, United States; [2]Department of Pathology, James J Peters VA Medical Center, Bronx, United States; [3]Office of Research and Development Veterans Health Administration, Washington, United States; [4]Department of Pathology University of Maryland School of Medicine, Baltimore, United States

**Abstract** Diagnosis of SARS-CoV-2 (COVID-19) requires confirmation by reverse transcription–polymerase chain reaction (RT–PCR). Abbott ID NOW provides fast results but has been criticized for low sensitivity. Here we determine the sensitivity of ID NOW in an ambulatory population presented for testing. The study enrolled 785 symptomatic patients, of whom 21 were positive by both ID NOW and RT–PCR, and 2 only by RT–PCR. All 189 asymptomatic patients tested negative. The positive percent agreement between the ID NOW assay and the RT–PCR assay was 91.3%, and negative percent agreement was 100%. The results from the current study were included into a larger systematic review of literature where at least 20 subjects were simultaneously tested using ID NOW and RT–PCR. The overall sensitivity for ID NOW assay was calculated at 84% (95% confidence interval 55–96%) and had the highest correlation to RT–PCR at viral loads most likely to be associated with transmissible infections.

*For correspondence: Jameel.iqbal@va.gov

## Introduction

The SARS-CoV-2 (COVID-19) virus has infected over 63 million people worldwide, causing over 1,500,000 deaths as of December 1, 2020. Infected individuals may be asymptomatic or may have a range of symptoms varying from a mild upper respiratory illness or gastrointestinal distress to severe respiratory distress with multisystem failure and death (*Wiersinga et al., 2020*). Definitive diagnosis requires laboratory detection of virus and is required for patients to be eligible for both clinical trials and current antiviral drugs and biologicals approved by the Food and Drug Administration (FDA) under Emergency Use Authorization (EUA) (*Tu and O'Leary, 2020*). Early in the pandemic, detection of SARS-CoV-2 relied predominately on reverse transcriptase–polymerase chain reaction (RT–PCR) assays performed in moderate to high complexity CLIA-certified laboratories. RT–PCR assays performed in certified laboratories are highly sensitive and specific, but require expensive and complex analyzers operated by certified and highly skilled laboratory workers; in many cases, these tests have required turnaround times of nearly a week or more.

The use of testing strategies with a rapid turnaround may allow for an earlier detection and better isolation of confirmed cases compared to laboratory-based diagnostic methods, as well as facilitate earlier treatment decisions and provide guidance on appropriate use of personal protective equipment. On March 27, 2020, Emergency Use Authorization was granted for the COVID-19 EUA assay on the ID NOW system (Abbott, Scarborough Diagnostics). The ID NOW system is a point-of-care (POC) device that uses an isothermal nucleic acid amplification technique to allow for nucleic acid amplification without thermal cyclers and allows for results to be obtained quickly. The ID NOW SARS-CoV-2 assay (Abbott) amplifies a unique region of the RdRp genome with a manufacturer's

claimed limit of detection (LOD) of 125 genome equivalents/mL. The isothermal technique allows for positive results to be available as soon as 5 min into the assay, and negative results within 13 min.

Since its release, several studies have been published demonstrating a sensitivity relative to RT–PCR from 44% to 94% (excluding a study with only a single positive). Studies have shown fairly definitively that the LOD for ID NOW COVID-19 requires significantly higher amounts of virus than most RT–PCR assays (*Smithgall et al., 2020*; *Zhen et al., 2020a*), but the clinical importance of this finding has been tempered by the observation that virus detectable only at high cycle time threshold (Ct) values is generally not culturable, and may therefore not be sufficiently high to infect others (*Mina et al., 2020*). Additional studies have suggested that nasal viral loads peak at around the time symptoms appear, and fall off as infection lingers (*Kucirka et al., 2020*). Hence, a diagnostic approach that is adequate early in the course of infection may be inadequate for patients that present later in the course of disease. Thus, the decision on whether the time advantage of a lower sensitivity device offsets the potentially higher LOD may depend on the context in which that device is employed.

To better understand the performance characteristics and trade-offs involved in the use of the ID NOW system, we have carried out a prospective clinical evaluation of the ID NOW system in the context of a community screening program focusing on symptomatic persons demonstrating one or more clinical feature of SARS-CoV-2 infection, comparing the results with those obtained by RT–PCR testing. We have augmented the findings of this investigation with a systematic review and meta-analysis of ID NOW performance, focusing on ambulatory community populations undergoing initial testing.

## Results

### Clinical evaluation

The evaluation enrolled 785 symptomatic patients, of whom 21 tested positive for SARS-CoV-2 by both the ID NOW and Hologic assays, and 2 tested positive only with the Hologic assay (*Table 1*). In addition, the evaluation enrolled 189 asymptomatic patients, none of whom tested positive by either ID NOW or RT–PCR. An 'invalid' ID NOW assay result was reported for nine subjects (two asymptomatic, seven symptomatic), all of whom tested negative by RT–PCR. Thus, the positive percent agreement between the ID NOW assay and the Hologic Panther Assay was 91.3%, and the negative percent agreement was 100%. The median cycle time (Ct) values in patients who had a positive Hologic RT–PCR was 28.2.

Two patients had discordant results with a negative ID NOW test and a positive Hologic RT–PCR test. The Hologic Ct values on the two discordant patients were 36.5 and 38.1. Of these discordant results, one patient is a 58-year-old woman who was a former smoker who presented with a cough and mild respiratory symptoms for approximately 6 weeks. She was retested 4 days after the initial discordant results at which time she tested negative in both the ID NOW and Hologic RT–PCR assays. The other patient with discordant results was a 34-year-old man with diabetes; he declined repeat testing but clinically was improving when contacted by phone.

**Table 1.** Results from the clinical evaluation comparing randomized anterior nares samples for the ID NOW compared to the Hologic Panther SARS-CoV-2 RT–PCR assay.

| | | Hologic result | | |
| --- | --- | --- | --- | --- |
| | | NEG | POS | Total |
| ID NOW result | NEG | 942 | 2 | 944 |
| | POS | 0 | 21 | 21 |
| | Invalid | 9 | 0 | 9 |
| | Total | 951 | 23 | 974 |
| Negative agreement | | 100.00% (82–100%) | | |
| Positive agreement | | | 91.30% (70–98%) | |

## Systematic review and meta-analysis

Forty papers were considered for inclusion. Of these, 14 met inclusion criteria, as reflected in the PRISMA diagram (*Figure 1*); 9 of those 14 studies enrolled 100 or more subjects. A brief summary of the studies included in our review, including the clinical study reported in this paper, is described in *Table 2*. A brief discussion of each paper including the results used in this review is presented in Appendix 1.

The risk of patient selection spectrum bias associated with the study population, or method of recruitment, was rated as either 'high' or 'unclear' for 12 of the published studies; this was the most

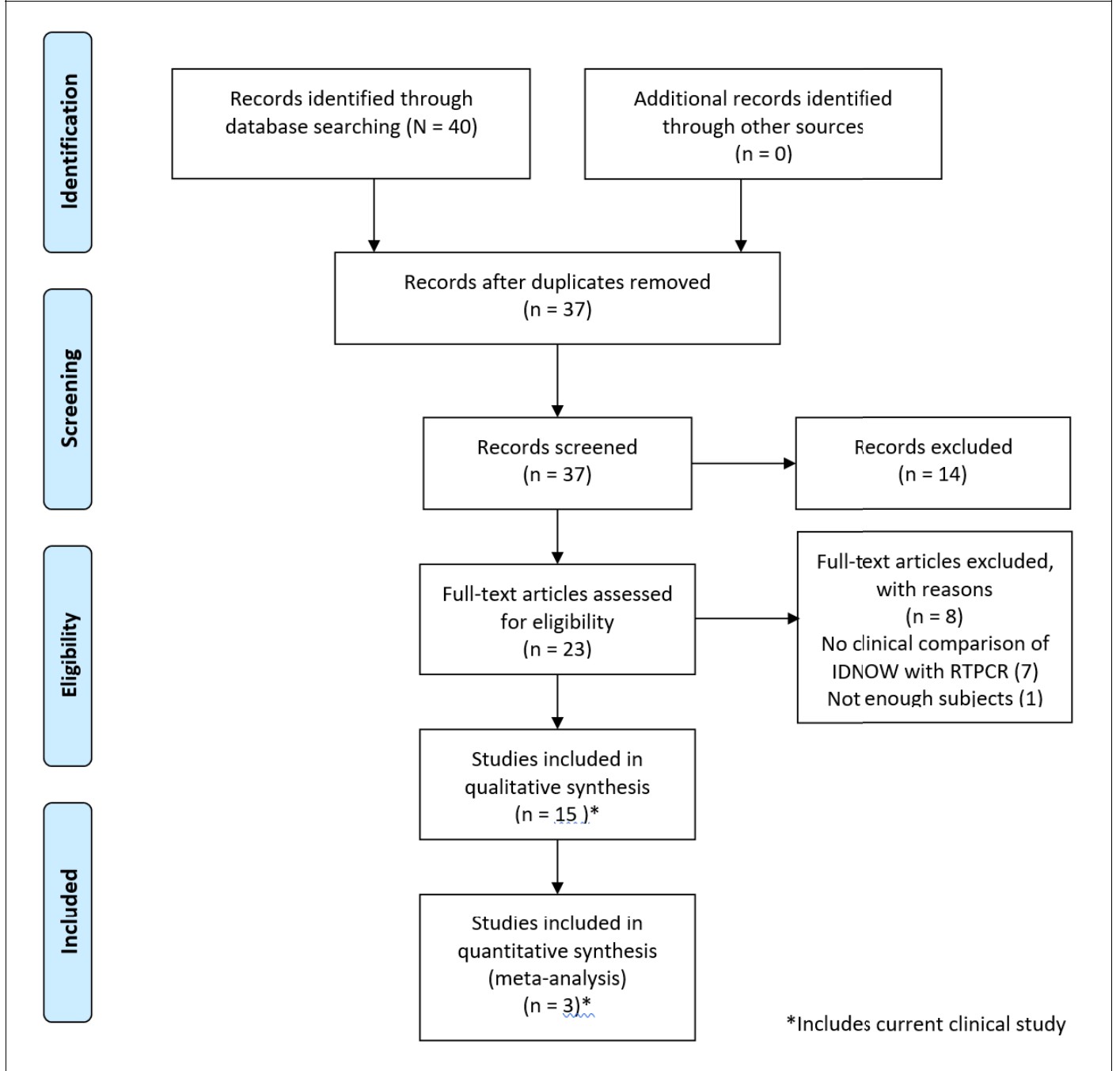

**Figure 1.** PRISMA 2009 flow diagram detailing the studies that were identified, screened, deemed eligible, and finally included in the analysis. Note that the data from the current clinical evaluation has been included in the analysis.

**Table 2.** Studies included in the systematic review.

| Study | Site | ID NOW sampling device | Residual sample tested? | ID NOW sample dry? | Timing of ID NOW test | Patient characteristics | Patient selection bias? | Index test bias? | Flow and timing bias | N (positive rate) | RT–PCR | Abbott ID now |
|---|---|---|---|---|---|---|---|---|---|---|---|---|
| | | | | | | | | | | | Positive/ CP % (95% CI) | Positive/ CP % (95% CI) |
| *Lephart et al., 2020* | AN | Foam | No | Yes | Up to 24 at 4°C | Convenience Hosp. and ED | Unclear | Unclear | Unclear | 88 (31%) | 27/27 91% (85–100%) | 12/27 44% (26–64%) |
| *Basu et al., 2020* | AN | Foam | No | Yes | Up to 24 at 4°C | Convenience Hosp. and ED | Unclear | Unclear | Unclear | 101 (31%) | 31/32 97% (82–100%) | 18/32 56% (38–73%) |
| *Cradic et al., 2020* | AN | Foam | No | Yes | Unclear | Symptomatic | Low | Unclear | Unclear | 182 (7.1%) | 13/13 100% (72–100%) | 12/13 92% (62–100%) |
| *Harrington et al., 2020* | AN | Foam | No | Yes | Unclear | Clinic, ED | Low | Unclear | Unclear | 524 (36%) | 186/188 99% (96–100%) | 141/188 75% (68–81%) |
| *Jin et al., 2020[†]* | U | U | No | Yes | Unclear | U | Unclear | Unclear | Unclear | 52 (12%) | 6/6 100% (52–100%) | 4/6 67% (24–94%) |
| *Mitchell and George, 2020* | NP | U | Yes | No | Unclear | U | High | High | Unclear | 61 (75%) | 46/46 100% (90–100%) | 33/46 72% (56–84%) |
| *Moore et al., 2020* | NP | U | No | No | Unclear | Symptomatic | High | Unclear | Unclear | 200 (64%) | 127/127 100% (96–100%) | 94/127 74% (65–81%) |
| *Rhoads et al., 2020* | AN, NP | U | Yes | No | Unclear | PCR+ | High | Unclear | Unclear | 96 (100%) | 96/96 100% (95–100%) | 90/96 94% (86–97%) |
| *Smithgall et al., 2020* | NP | U | Yes | No | Up to 48 hr at 4°C | Selected | High | Unclear | Unclear | 113 (80%) | 89/90 99% (94–100%) | 65/88 74% (64–82%) |
| *Thwe and Ren, 2020* | NP | U | No | Yes | ~2 hr | Symptomatic Hosp. and ED | High | Unclear | Unclear | 129 (7.7%) | 15/15 100% (75–100%) | 8/15 53% (27–78%) |
| *Zhen et al., 2020b* | NP | Many | Yes (most) | No | Up to 72 hr at 4°C | Symptomatic | Unclear | Unclear | Unclear | 107 (54%) | 57/58 98% (90–100%) | 50/57 88 (76–95) |
| *Comer and Fisk, 2020* | AN | Foam | No | Yes | 15 m | U | High | Low | Low | 117 (0.9%) | 1/1 100% (5–100%) | 0/1 0% (0–95%) |
| *Ghofrani et al., 2020* | AN | Many | No | Yes | Unclear | Mostly symptomatic convenience | High | Unclear | High | 113 (16%) | 17/18 94% (71–99%) | 17/18 94% (71–99%) |
| *SoRelle et al., 2020[‡]* | S | Tube | No | No | Unclear | U | Unclear | Low | Low | 59 (39%) | 23/23 100% (82–100%) | 18/23 78% (56–92%) |

*Table 2 continued on next page*

*Table 2 continued*

| Study | Site | ID NOW sampling device | Residual sample tested? | ID NOW sample dry? | Timing of ID NOW test | Patient characteristics | Patient selection bias? | Index test bias? | Flow and timing bias | N (positive rate) | RT–PCR | Abbott ID now |
|---|---|---|---|---|---|---|---|---|---|---|---|---|
| Tu (this study) | AN | Foam | No | Yes | 15 m | symptomatic and asymptomatic | Low | Low | Low | 974 (2.4%) | 23/23 100% (82–100%) | 21/23 91% (70–98%) |

*Likely <2 hr in some cases, more than a single type of sampling device was used. When a dry ID NOW foam swab was used as a part of this study, both the table above and the results reflect the use of that device, which is consistent with the current ID NOW package insert. Comparisons based upon use of other transport media are only shown when no data was presented for use of dry swabs.

†Table shows only the comparison between ID NOW and Cobas using dry swabs.

‡Table shows comparison of saliva tested on ID NOW vs saliva tested using Cepheid Xpert Xpress SARS-CoV-2.

Abbreviations in table: AN = anterior nares; OP = oropharynx; NP = nasopharynx; S = saliva; U = unknown.

Data are only presented from papers in which it was possible to construct a composite 'gold standard' in which a positive result on any platform contributed to create a 'composite positive (CP)'. Specificity was assumed to be 100% for all platforms/tests. This differs from the method presented in some of the papers incorporated into this table.

common concern raised in the quality assessment. Studies with a high or unclear risk of bias were characterized by failure to present patient symptom status (five studies), inclusion of subjects who had previously tested positive for SARS-CoV-2 (one study) or use of investigator-selected or non-clinical convenience samples. Evidence of bias associated with the conduct of RT–PCR testing was not identified for any of the 14 studies meeting inclusion criteria. Several studies suffered either from unclear or elevated risk of index test or from flow and timing biases (detailed further in Appendix 1).

The clinical sensitivity of the ID NOW assay was lower than that of the RT–PCR assay, when both were compared to the composite reference standard, in 14 of the 15 studies shown in *Table 2*. In studies reporting more than a single positive RT–PCR result, the sensitivity of ID NOW, as compared to the composite reference standard, varied from 44 to 94%, while that of the RT–PCR test varied from 91 to 100%. For studies in which patient selection bias was rated low, the sensitivity of ID NOW (in comparison with the composite reference standard) ranged from 60 to 92% (*Table 2*). This corresponds with published analytical sensitivity estimates that have shown limits of detection for ID NOW that are several orders of magnitude higher than those of RT–PCR assays, ranging from 3900 (*Lephart et al., 2020*) to 20,000 (*Zhen et al., 2020a*) gene copies/mL, and data published on an FDA web site (https://www.fda.gov/medical-devices/coronavirus-covid-19-and-medical-devices/sars-cov-2-reference-panel-comparative-data) that suggests a 500-fold higher LOD for the ID NOW platform than for the Panther Fusion Assay employed in our clinical study. These results are consistent with the studies in our systematic review that showed discordance among assays to be most frequent when Ct values were relatively high (see Appendix 1) (*Basu et al., 2020*; *Cradic et al., 2020*; *Lephart et al., 2020*; *Mitchell and George, 2020*; *Smithgall et al., 2020*; *Zhen et al., 2020a*).

The ID NOW instructions for use (IFU, https://www.fda.gov/media/136525/download) have changed over time, but generally have called for samples to be tested no later than 1 hr after specimen acquisition and kept at room temperature during that period. The changes in the IFU have made it difficult to assess whether published studies provided sufficient information to allow a determination that conformation to instructions for use was followed sufficiently. Four studies included in this review were based upon a split/residual sample design; the calculated sensitivity for the ID NOW in these studies ranged from 72% to 94%. For eight of the studies, timing of the ID NOW test was unclear, while for four studies, samples were held after collection at 4°C for up to 24 hr (two studies), 48 hr (one study), or 72 hr (one study). The degree to which this affects assay sensitivity is unclear; however, it is noteworthy that a study that held samples for up to 72 hr reported ID NOW sensitivity (as compared to the composite reference standard) of 88%, while another study that held samples for no more than 2 hr reported a sensitivity of 56%. Only one of the studies captured for this systematic review reported a time-to-test for ID NOW of ≤1 hr, and that study included only one patient that tested positive using either device. Thus, there is no conclusive evidence that the refrigeration serves as an explanation for varying sensitivities.

There was no obvious relationship between the sample site, such as anterior nares (AN) versus NP, or sampling device and the sensitivity of the ID NOW test. Both high and low concordance with the composite reference standard were found for both sites and for both foam and flocked swabs. Similarly, both good performance and poor performance were found for both samples transported in a medium or transported dry. Finally, the overall prevalence of positive findings in the study population was not correlated with the performance of ID NOW in the studies we have examined.

We included the two cohorts with low risk of patient selection bias, together with the current study, in a meta-analysis, the results of which are shown as forest plots in *Figure 2*. The sensitivity of ID NOW, as compared with the reference standard, was estimated at 82% (*Figure 3A*); the lower and upper 95% confidence bounds were 67% and 91%, respectively. Measures of heterogeneity did not reach statistical significance ($\tau^2$ = 0.25, Q[df = 2]=3.67, p=0.16, $I^2$ = 45.53). In contrast, the sensitivity of RT–PCR (*Figure 3B*) was estimated at 98% with a 95% confidence interval (CI) of 96–99%. There was no suggestion of heterogeneity ($\tau^2$ = 0.000, Q[df = 2] = 0.453, p=0.112, $I^2$ = 0.000). The sensitivity estimates for both ID NOW and RT–PCR were reduced, probably by about 2%, by the need to include a continuity correction in the der Simionian–Laird computations.

## Discussion

We conducted a large clinical evaluation of the ID NOW isothermal PCR system in a low-prevalence population and found that the ID NOW system had a positive percent agreement of 91% and a

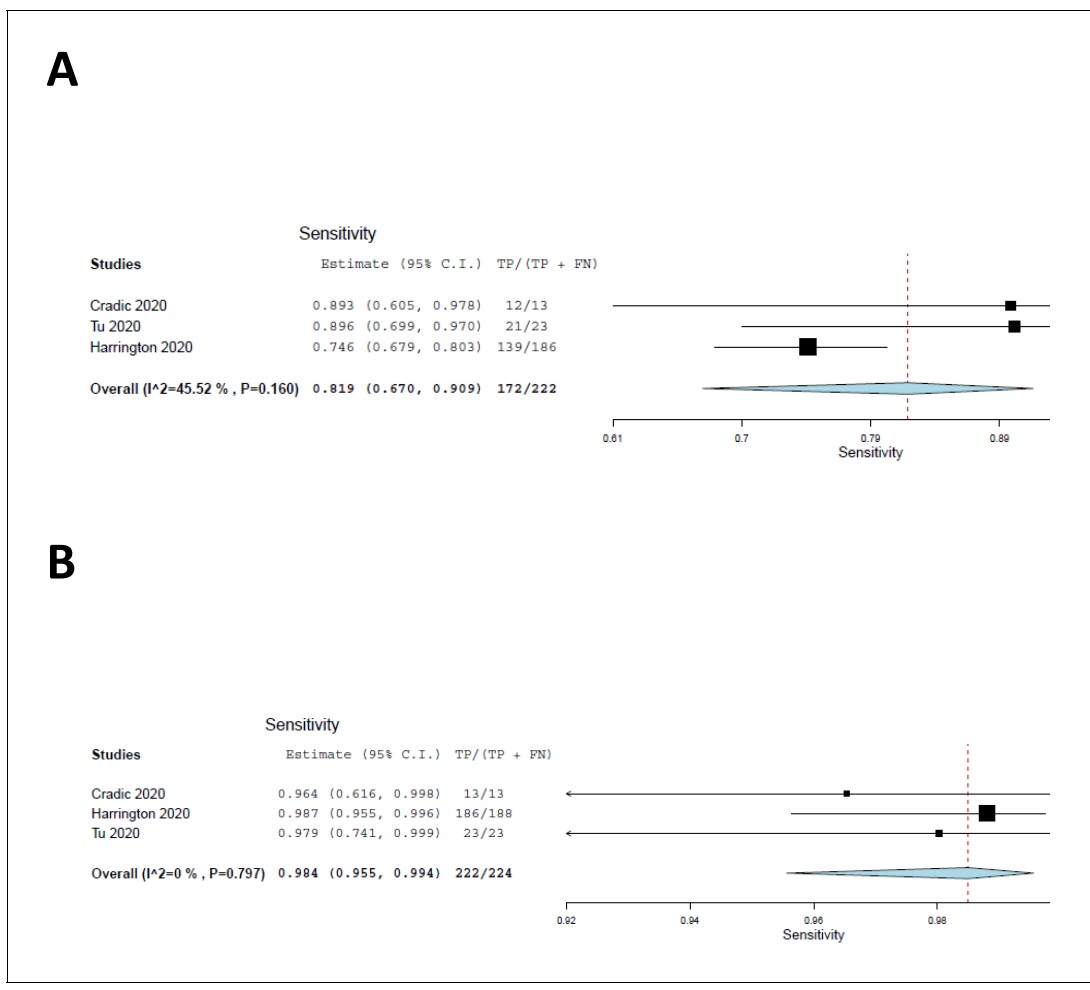

**Figure 2.** Forest plots demonstrating the three studies with low risk of patient selection bias utilized in the meta-analysis. (**A**) The sensitivity of ID NOW as compared with the reference standard, and the overall sensitivity was estimated to be 82% with a lower 95% confidence bound at 67% and an upper bound of 91%. (**B**) The sensitivity of RT–PCR and is estimated to be 98% with a 95% CI of 96–99%.

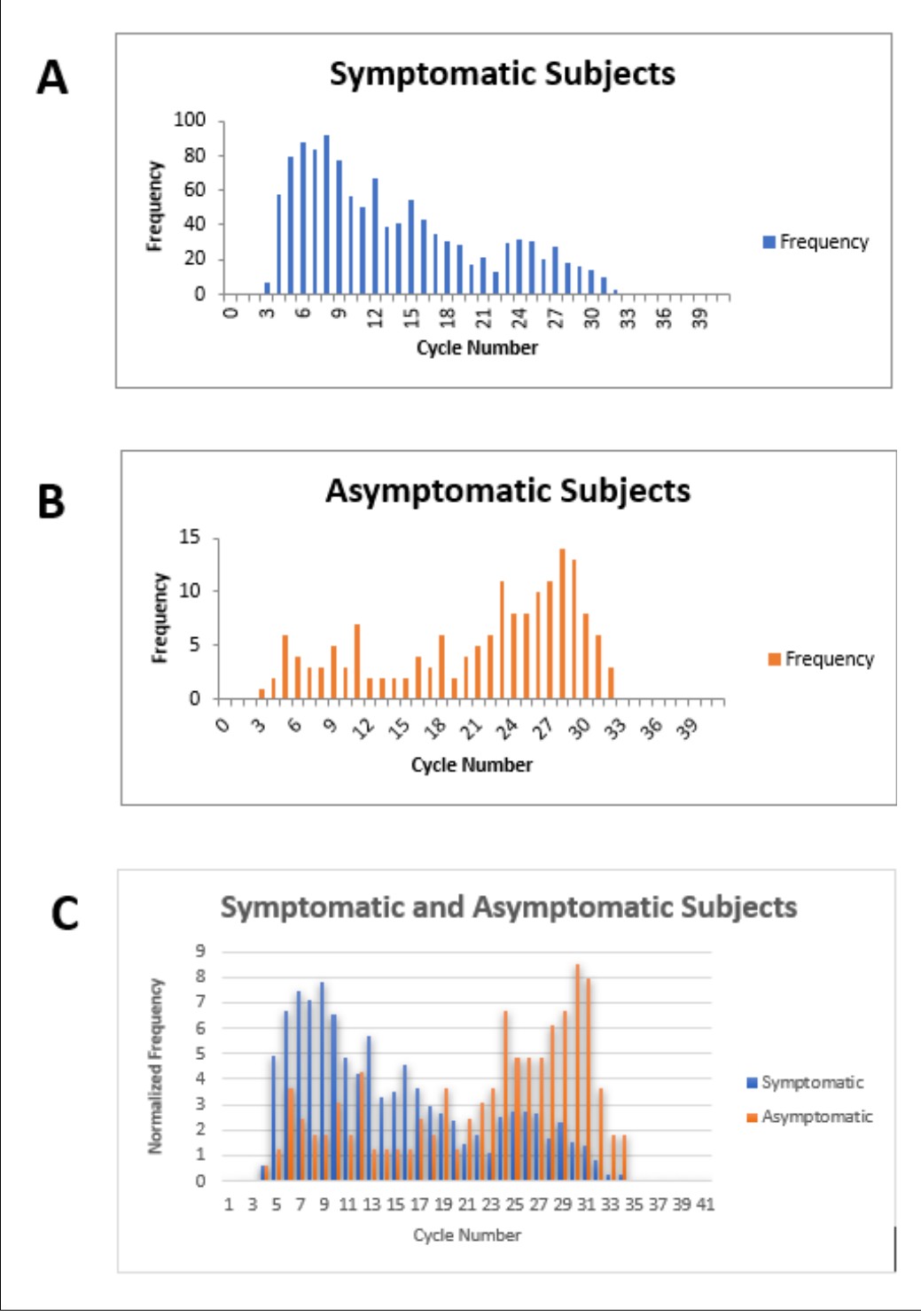

**Figure 3.** Cycle number distributions for 1182 symptomatic (**A**) and 164 asymptomatic (**B**) patients who tested positive for SARS-CoV-2 between July 14 and November 16, 2020, using the Abbott m2000 assay at The Everett Clinic. For patients with multiple tests, only the first positive test is included. In (**C**), data for each group of patients has been normalized so that the sum of all bins is 100, allowing better comparison of the distributions. The Abbott m2000 cycle number is generally about 10 cycles less than the Ct reported for PCR assays on other devices.

negative percent agreement of 100% compared to the Hologic Panther RT–PCR system. Several features that distinguish this study from those included in the systematic review are worth noting. The first is that the time from specimen collection to ID NOW testing was 15 min or less for most individuals tested. None of the studies meeting criteria for inclusion in the systematic review had such a short collection-to-testing time. This may account, at least in part, for the relatively high positive percent agreement found in our studies by comparison with most previous reports (although we note overlap of CIs for studies included in the meta-analysis, as shown in *Figure 2* and *Table 2*). A second feature of the current clinical evaluation, shared by only two of the studies included in our systematic review, was that the sample was based on a subject group that resulted from an attempt to enroll virtually every patient who walked through the door. Our study did not find cases in which NP specimens tested by RT–PCR were negative in the face of a positive ID NOW result; this finding is similar to the findings of our systematic review, which found only four such cases among 1942 tests, as seen in Appendix 1.

Most of the variation in performance reported for the ID NOW system seems to result from the differences in recruitment strategies employed in these studies. Peak viral loads and transmission risk for SARS-CoV-2 are found in symptomatic patients at symptom onset and then fall throughout the course of disease. Because RT–PCR assays have a LOD that are several orders of magnitude lower than that of the isothermal PCR ID NOW assay, one would expect them to remain positive for significantly longer times after the time of peak viral load. The use of 'convenience samples', particularly populations including patients who have been hospitalized after a diagnosis of COVID-19, may include more patients who are past their period of peak viral load compared to a sample of ambulatory patients first presenting for evaluation – such as those in our study who appeared for testing because of recent symptom onset. The two studies that met inclusion criteria for our review, which had the lowest positive percent agreement between ID NOW and RT–PCR, included hospitalized patients (*Lephart et al., 2020*; *Thwe and Ren, 2020*) although another study with very low concordance did not (*Basu et al., 2020*).

The conclusions from our clinical study are limited by a relatively small number of positive cases; nonetheless, the high level of agreement with RT–PCR suggests that ID NOW is effective at identifying, or excluding, SARS-CoV-2 in a symptomatic ambulatory patient population. The systematic review and meta-analysis generally support this conclusion, although they suggest a reduction in sensitivity NOW, in comparison with that of RT–PCR, that may be clinically significant under some circumstances. The specificity of a positive ID NOW result appears to be upwards of 99.8%, based upon the studies included in this review.

Under the conditions of the current clinical study (population prevalence of 2.36%), the positive and negative predictive values of the ID NOW test were 100% and 99.8% (99.2–99.9%), respectively. At a prevalence of 10% in the tested population, the positive and negative predictive values are 100% and 99% (96.49–99.74%), respectively. Using the 82% estimate from our meta-analysis in a 2% positive population yields a negative predictive value of 99.6% (99.4–99.7%), which drops to 98.0% (97.1–98.7%) in a population with 10% disease prevalence. At the lower 95% confidence limit of the meta-analysis (67%), negative predictive value remains acceptable at 99.2% (99–99.4%) for a population prevalence of 2.3%. It becomes more marginal at 96.5% (95.4–97.3%) when the prevalence of disease in the tested population goes to 10% or higher.

Our clinical study and meta-analysis suffer from several limitations. The data from our clinical study does not provide information on the potential utility of ID NOW in testing an asymptomatic patient population, since no positive cases were identified among the enrolled asymptomatic patients similarly, our systematic review and meta-analysis does not focus on this group. Comparison of RT–PCR cycle numbers between symptomatic and asymptomatic ambulatory outpatients from The Everett Clinic suggests that the viral load for symptomatic patients is generally higher than for asymptomatic patients (*Figure 3*). This observation, which has also been reported elsewhere (*Ra et al., 2021*), raises the possibility that ID NOW may miss infections in the asymptomatic infected population. On the other hand, the observation that specimens that demonstrate high Ct values are unlikely to be successfully cultured raises the possibility that many of these patients are less likely to transmit the infection, although the relationship between the ability to culture virus and infectivity has yet to be demonstrated for SARS-CoV-2. Our clinical study also suffered a significant loss of power to assess ID NOW sensitivity as a result of the low number of positive results, and the reduction of sample size caused by the decision to terminate the study as a result. The meta-analysis

is also limited by the small number of studies meeting inclusion criteria, and the fact that positive cases are heavily concentrated in only a single study. Strengths of the clinical study include pre-trial power analysis with sample size estimation, precise adherence to the ID NOW specimen acquisition protocol, and extremely high power for assessing assay specificity. Taken together with the focus on initial diagnosis of disease in the studies included in the meta-analysis, we believe the combination of trial and meta-analysis provides useful information for clinicians for whom POC testing is helpful.

POC testing has substantial advantages over laboratory-based testing when a patient presents with symptoms characteristic of COVID-19. Patients who are SARS-CoV-2 positive can be asked to isolate immediately, and patients who test negative can be reassured or retested using a more sensitive test, depending on clinical judgment. Although the performance of ID NOW in an asymptomatic population has not been established, and caution may be appropriate when using ID NOW with a high-risk population, increased frequencies of testing, together with a rapid turnaround time, are likely to have greater impact on population health outcomes than are differences in test sensitivity (*Larremore et al., 2021*; *Mina et al., 2020*). In addition, the ID NOW system provides excellent negative predictive value in symptomatic ambulatory patients, particularly when the population prevalence of SARS-CoV2 is low. It thus provides a speedy and effective alternative to laboratory-based RT–PCR methods under many clinical circumstances.

# Materials and methods

**Key resources table**

| Reagent type (species) or resource | Designation | Source or reference | Identifiers | Additional information |
|---|---|---|---|---|
| Commercial assay, kit | ID NOW SARS-CoV-2 reagents | Abbott | Cat. # 190–000 for COVID reagents Cat. # 190–080 for COVID controls | https://www.fda.gov/media/136525/download |
| Commercial assay, kit | SARS-CoV-2 reagents | Hologic | Cat. # PRD-06420 for the COVID reagents and Cat. # 303014 for the assay fluids reagents | https://www.hologic.com/sites/default/files/2020-09/AW-21159-001_004_01.pdf |

## Study population and sample collection
### Clinical study
The IRB-approved clinical study was conducted at The Everett Clinic between April 8 and 22, 2020, and engaged ambulatory symptomatic patients seen in the febrile upper respiratory infection (F/URI) clinics and other patients from non-F/URI clinics. Patients who were unable to demonstrate understanding of the study, not willing to commit to having all samples collected, had a history of nosebleed in the past 24 hr, nasal surgery in the past 2 weeks, chemotherapy treatment with documented low platelet and low white blood cell counts, or acute facial trauma were excluded; nonetheless, an attempt was made to consecutively enroll all eligible patients.

The original study design called for enrolling 2000 symptomatic and 500 asymptomatic subjects, which would have provided, in the symptomatic population, power of 80% for finding a difference (at $\alpha = 0.05$) of 5% in the sensitivity of ID NOW compared with the RT–PCR reference standard; inclusion of at least 1350 negative patients would have provided 95% power (at $\alpha = 0.025$) for finding a 5% difference in specificity. The study design assumed a population prevalence of 10%, and the study was terminated early when the population prevalence dropped to such a low level as to make the study unaffordable. We have re-estimated the power of this study without reference to the observed results but considering the sample size and proportion of RT–PCR-positive tests that were observed when the study was terminated. This re-estimation suggests that the study retained 80% power to find a difference of 15% or more in sensitivity between ID NOW and RT–PCR, and well over 95% power to find a difference in specificity of more than 5%. Indeed, the significant drop in population prevalence that led to a loss of power for detecting loss of sensitivity resulted, as expected (*Bujang, 2016*), in an increase in power for detecting loss of specificity.

Patients who consented to the study had two sterile foam swabs (Puritan, #PK002196) obtained by trained clinical staff. To ensure maximum loading of viral material, each swab sampled in both AN. To ensure that both swabs had equal opportunity to collect viral material (*Figure 4*), the collection of the two swabs used a cross-over method.

The procedure is as follows:

1. The first swab was gently inserted into the right nostril until resistance was met at the level of the turbinate (less than one inch into the nostril), and gentle pressure was applied to the outside nasal wall and the swab was rotated several times against the nasal wall and then slowly remove from the nostril.
2. The second swab was gently inserted into the left nostril, and sampling was obtain in a similar manner.
3. Next, the first swab was inserted into the left nostril, and sampling was obtained in a similar manner.
4. Finally, the second swab was inserted into the right nostril, and sampling was obtained in a similar manner.

If the patient's year of birth ended in an even year, the first swab inserted into the right nostril was designated for SARS-CoV-2 testing using the POC analyzer. If the patient's year of birth was an odd year, the first swab inserted into the left nostril was designated for SARS-CoV-2 testing using the ID NOW analyzer. The swab designated for testing in the ID NOW analyzer was reinserted into the original paper sleeve packaging, a patient label was affixed, placed in a plastic bag, and transported to the clinic lab on site for immediate testing. Typically, fewer than 15 min passed between the time the room-temperature sample was collected and the time that the swab was inserted into the ID NOW sample receiver.

The remaining swab was placed in VTM (Medical Diagnostic Laboratories, LLC). After a patient label was affixed, the specimen was placed in a plastic bag and transported at 4°C to the University of Washington Virology Lab, where a Hologic Panther Fusion SARS-CoV-2 assay (Marlborough, MA) was performed per manufacturer's recommendations. With the Hologic assay, a sample is

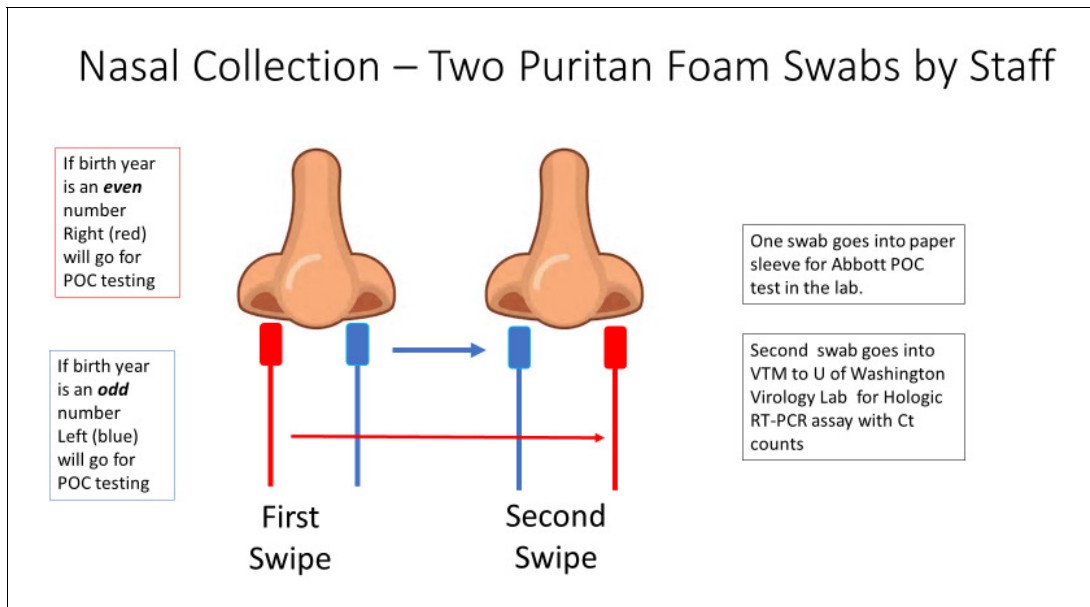

**Figure 4.** The collection methodology to ensure proper randomized sampling of nares for simultaneous analysis for SARS-CoV-2 by the ID NOW isothermal amplification and Hologic Panther RT–PCR assays. A total of two swabs was collected on each patient, with patients having an even birth year number the right nares were collected first followed by a second swipe in the left nares and then for ID NOW point-of-care (POC) testing (depicted as red swab). For those patients, the other swab (blue swab) was sent for SARS-CoV-2 RT–PCR analysis by the Hologic Panther assay. For patients having an even birth year, the swabs sent for testing was reversed with the blue swab sent for ID NOW testing and the red swab sent for RT–PCR analysis.

considered positive if an amplification signal is detected at a cycle time (Ct) of 42 cycles or less. Those involved in the RT–PCR assay were blinded to the ID NOW result.

ID NOW results, which were reported as 'invalid', were treated as negative when calculating the sensitivity of the ID NOW test; moreover, they were excluded from computations of specificity since this result would be expected to trigger reflex testing. Confidence intervals for sensitivity and specificity were calculated using *Newcombe, 1998* efficient score method (with continuity correction) as implemented in the Vassarstats calculator for confidence intervals of a proportion (http://vassarstats. net/).

The human ethics review and IRB for these studies was approved by the United Health Group Office of Human Research Affairs (OHRA), Federal wide Assurance #: FWA00028881, OHRP Registration #: IORG0010356.

## Systematic review and meta-analysis

Our systematic review was designed to answer two questions:

> What is the LOD for the ID NOW assay?
> What is the clinical sensitivity of the ID NOW SARS-CoV-2 assay in comparison with an RT–PCR assays for SARS-CoV-2?

The study is based on a protocol registered on PROSPERO (CRD42020204441), but a complete protocol has not been published. PubMed, medRxiv, and bioRxiv were searched over the interval from January 1, 2020 to August 16, 2020, using the search 'ID NOW', 'isothermal amplification', and 'lamp isothermal'. Following the initial identification of papers, the titles and abstracts were screened to eliminate papers not meeting the prespecified inclusion criteria as defined below and diagramed in *Figure 2*. Papers remaining after this process were rescreened, particularly since many of the papers reviewed were in the form of research letters that did not have an abstract. Ultimately, 14 papers that met inclusion criteria for clinical comparison were available for analysis, as shown in the PRISMA flow diagram (*Figure 2*). In addition, one additional paper addressing the LOD for ID NOW was identified (*Fung et al., 2020*).

To be included in the systematic review, studies were required to include a minimum of 20 unique subjects. Studies must have compared samples obtained simultaneously from the same site or from an equivalent site. Both split-sample designs and independent sample designs were considered. Results must have been reported in a manner that allowed construction a confusion matrix including the RT–PCR and ID NOW test. Because 'discrepant analysis' provides biased sensitivity estimates, studies using this technique to resolve diagnostic conflicts between two sites were not to be included unless data could be analyzed independently of the discrepant analysis. If multiple time points were included in one of the included studies, only the first time point was to be used in our analysis. If confusion matrices could only be constructed from data involving multiple time points from the same patients, the study was excluded. No attempt was made to obtain data from the investigators involved in these published studies.

Study information was recorded on a predetermined data extraction form that included study author, type of study, inclusion and exclusion criteria, setting, sample types, swab types, transport medium, manufacturer or description of nucleic acid amplification assays, as well as space to record study results in the form of confusion matrices. The potential for bias associated with each study was evaluated using the QUADAS2 instrument. The risk of spectrum bias, which is the variability of medical test performance that happens when tests are given to different mixes of patients at different locations, was assessed from the perspective of testing as an initial diagnostic method; the risk estimate does not constitute a judgment on the quality of the study, which may have been performed to demonstrate assay validity, assessment of recovery, or other purposes different than that for which we evaluated potential bias.

Because the choice of any particular diagnostic device as a 'gold standard' provides a biased estimate of relative sensitivity, which compared with all other devices, a composite reference standard (CRS) was computed for each study on the basis of all devices and sample types included in the study, when possible. Equivocal results and assay failures were not used in the calculation of sensitivity or in the construction of the CRS for each study. Where multiple RT–PCR assays were performed, only the performance of the most sensitive of these assays (as measured using the composite reference standard) is reported in results tables. Confidence limits for sensitivity were computed using

Newcombe's efficient score method, as above. Criteria for performing a formal meta-analysis were prespecified as follows: (1) studies used the same amplification technology (such as RT–PCR) as a reference; (2) studies used the same upper airway sample site (AN, mid-turbinate [MT], and NP could be included together, but not admixed with studies based on oropharynx samples); (3) studies enrolled a similar patient mix (e.g. symptomatic, asymptomatic, hospitalized and similar clinical environment [drive-through/community health center or hospital]). Three papers in which with a low risk of bias were deemed appropriate to include in a meta-analysis were analyzed using a diagnostic effects model (der Simion–Laird) as implemented by OpenMetaAnalyst software program.

The choice of any particular diagnostic device as a 'gold standard' provides a biased estimate of relative sensitivity which compared with all other devices (*Baughman et al., 2008*). When two devices, each of which is expected to have a near-zero false positive rate, are being compared, the use of a CRS is a reasonable approach by which to reduce this bias (*Tang et al., 2018*). For this reason, we compared the performance of ID NOW and RT–PCR methods with a composite reference standard in which the specificity of all assays was considered to be perfect, and a positive result for any assay was considered to be a 'true positive'. Equivocal results and assay failures were not used in the calculation of sensitivity or in the construction of the CRS for each study. Where multiple RT–PCR assays were performed, only the performance of the most sensitive of these assays (as measured using the composite reference standard) is reported in results tables. Confidence limits for sensitivity were computed using Newcombe's efficient score method, as above. Criteria for performing a formal meta-analysis were prespecified as follows: (1) studies used the same amplification technology (such as RT–PCR) as a reference; (2) studies used the same upper airway sample site (AN, MT, and NP could be included together, but not admixed with studies based on OP samples); (3) studies enrolled a similar patient mix (e.g. symptomatic, asymptomatic, hospitalized) and similar clinical environment (drive-through/community health center or hospital). Three papers in which with a low risk of bias were deemed appropriate to include in a meta-analysis were analyzed using a diagnostic effects model (*DerSimonian and Laird, 1986*) as implemented by OpenMetaAnalyst software program (*Wallace et al., 2012*). Since our model is built on the assumption that there are no false positive ID NOW results, a value of 0.5 was added to all cells as a continuity correction.

## Acknowledgements

The authors wish to acknowledge the contributions of Garrett Galbreath, Lorraine Bell, Susan Spanos, Anne Hartman, Amanda Wells, Kim Gangloff, Jeremy Norris, and Korie Packwood (from The Everett Clinic Laboratory, analytical and electronic medical records staff), The Everett Clinic nurses, medical assistants, staff, providers and patients who graciously contributed to this study. Funding: Funds for the clinical studies, Protocol Number 2008201 and 2008202, and Clinical Evaluation of the ID NOW COVID-19 Assay in Symptomatic and Asymptomatic Subjects were provided to YT and The Everett Clinic by Abbott Scarborough Diagnostics. JI is grateful to the National Institutes of Health for grant support, namely U19 AG60917 and R01 DK113627.

## Additional information

### Competing interests

Jameel Iqbal: Reviewing editor, *eLife*. The other authors declare that no competing interests exist.

### Funding

| Funder | Grant reference number | Author |
| --- | --- | --- |
| NIH Clinical Center | U19 AG60917 | Jameel Iqbal |
| NIH Clinical Center | R01 DK113627 | Jameel Iqbal |
| NIH Clinical Center | 2008201 | Yuan-Po Tu |
| NIH Clinical Center | 2008202 | Yuan-Po Tu |

The funders had no role in study design, data collection and interpretation, or the decision to submit the work for publication.

## Author contributions
Yuan-Po Tu, Conceptualization, Resources, Data curation, Formal analysis, Supervision, Funding acquisition, Validation, Investigation, Writing - original draft, Project administration, Writing - review and editing; Jameel Iqbal, Conceptualization, Formal analysis, Validation, Writing - original draft, Project administration, Writing - review and editing; Timothy O'Leary, Conceptualization, Data curation, Software, Formal analysis, Supervision, Validation, Investigation, Visualization, Methodology, Writing - original draft, Project administration, Writing - review and editing

## Author ORCIDs
Jameel Iqbal ⓘ https://orcid.org/0000-0002-4598-5064
Timothy O'Leary ⓘ https://orcid.org/0000-0002-2435-9136

## Ethics
Human subjects: The human ethics review and IRB for these studies was approved by the United Health Group Office of Human Research Affairs (OHRA), Federal wide Assurance #: FWA00028881, OHRP Registration #: IORG0010356.

## Decision letter and Author response
Decision letter https://doi.org/10.7554/eLife.65726.sa1
Author response https://doi.org/10.7554/eLife.65726.sa2

## Additional files
### Supplementary files
• Transparent reporting form

### Data availability
All data used for analysis has been included in the figures, tables and two appendices.

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

## Appendix 1

## Brief descriptions and confusion matrixes for each study

Computations using multiple samples to compute the composite reference standard are not shown.

### Lephart et al., 2020

This paper reports a comparison of Abbott ID NOW, Abbott m2000, Diasorin Simplexa and Cepheid Xpert Xpress using 75 nasopharyngeal (NP) and anterior nares (AN) swabs obtained from patients presenting in the ED and 13 from recovering COVID-positive inpatients. NP swabs were transported to a central laboratory and tested with the Simplexa, following which residual specimens were tested within 24 hr on the m2000 and Xpert Xpress devices. Nasal swabs were collected in parallel, transported dry to the laboratory, stored at 4°C, and tested by ID NOW within 24 hr. The potential for patient selection bias is unclear because of the inclusion of recovering hospitalized subjects. Storage at 4°C for up to 24 hr is not consistent with current Abbott ID NOW instructions for use (IFU), resulting in unclear, though possibly insignificant risks for index test and flow and timing biases.

Data in the confusion matrix below (*Appendix 1—table 1*), and in *Table 2*, reflects comparison of ID NOW with the Cepheid Xpert Xpress assay. Investigators noted that positive agreement was higher in patients with low m2000 cycle numbers.

Investigators also determined the limits of detection (LOD) for ID NOW and the m2000 assay, finding a LOD of 262 copies/mL for ID NOW, and 32.5 copies/mL for the m2000.

Appendix 1—table 1. *Lephart et al., 2020* Confusion matrix.

|  |  | ID NOW | |
|  |  | Positive | Negative |
| --- | --- | --- | --- |
| Cepheid XpertXpress | Positive | 12 | 15 |
|  | Negative | 0 | 60 |

### Basu et al., 2020

The investigators obtained 101 paired foam AN (both nares) and NP swabs from patients presenting to the emergency department of a New York City Hospital between April 22 and 24, 2020 (*Appendix 1—table 2*). All swabs were transported to the laboratory at room temperature, while NP swabs were transported in VTM. The dry nasal swabs were tested within 2 hr of collection or kept at 4–8°C for up to 24 hr before testing on the ID NOW platform. NP swabs were tested on the Cepheid Xpert Xpress. Since no information is given about recruitment strategy, the risk of patient selection bias is unclear. Storage at 4–8°C for up to 24 hr is not consistent with current Abbott ID NOW instructions for use, resulting in unclear, though possibly insignificant risks for index test and flow and timing biases.

Investigators noted that all six patients with Xpert Xpress N2 Ct value of 33.5 tested positive by ID Now, while the positive percent agreement dropped substantially for cases in which the Xpert Xpress Ct value was higher.

Appendix 1—table 2. *Basu et al., 2020* Confusion matrix.

|  |  | ID NOW | |
|  |  | Positive | Negative |
| --- | --- | --- | --- |
| Cepheid Xpert Xpress | Positive | 17 | 14 |
|  | Negative | 1 | 69 |

### Cradic et al., 2020

NP, oropharyngeal (OP), and AN swabs were obtained prospectively from 182 consenting patients seen in an emergency department (*Appendix 1—table 3*). NP swabs were placed in viral transport

media (VTM) and transported to the lab for testing both by Diasorin Simplexa and Abbott ID NOW. With AN and OP, swabs were tested by ID NOW. Risk of patient selection bias is low, but there is a lack of information regarding specimen flow and timing; thus, the risk of index test bias, and flow and timing bias is unclear.

Investigators used serial dilution (in VTM) of patient specimens to assess relative sensitivity of ID NOW compared with Roche Cobas and Diasorin Simplexa; the results suggest that ID NOW has a limit of detection about 10-fold higher than that of the Diasorin assay, and 100-fold higher than that of the Roche assay.

Appendix 1—table 3. *Cradic et al., 2020* Confusion matrix.

|  |  | ID NOW | |
|  |  | Positive | Negative |
| --- | --- | --- | --- |
| Diasorin Simplexa | Positive | 12 | 1 |
|  | Negative | 0 | 169 |

## Harrington et al., 2020

Paired foam nasal swabs (NS) and NP swabs were obtained from 524 symptomatic subjects presenting consecutively at three emergency departments (ED) and two immediate care centers. Both nasal swabs (dry) were tested locally using ID NOW, and NPS were transported to a central laboratory and NPS (in VTM) were transported to a central laboratory and tested using the Abbott m2000 (*Appendix 1—table 4*). The risk of patient selection bias was rated to be low; since no information was given regarding the interval between specimen acquisition and ID NOW testing, the risk of index test bias and flow and timing bias were rated unclear.

Harrington estimated the LOD for the ID NOW assay to be 3225 copies/mL, based on the package insert which reports an LOD of 100 genome equivalents/mL.

Appendix 1—table 4. *Harrington et al., 2020* Confusion matrix.

|  |  | ID NOW | |
|  |  | Positive | Negative |
| --- | --- | --- | --- |
| Abbott m2000 | Positive | 139 | 47 |
|  | Negative | 2 | 336 |

## Jin et al., 2020

As part of a larger study, investigators compared the results from 52 dry swabs tested with the ID NOW system with those of paired NP specimens tested on the Roche COBAS system. Details regarding patient recruitment and the patient to test time for were not provided (*Appendix 1—table 5*). For this reason, the risk of patient recruitment bias, index test bias and flow and timing bias are all considered to be unclear.

Investigators also estimated that the LOD for the ID NOW assay is 16-fold higher than that of the Roche Cobas assay.

Appendix 1—table 5. *Jin et al., 2020* Confusion matrix.

|  |  | ID NOW | |
|  |  | Positive | Negative |
| --- | --- | --- | --- |
| Roche COBAS | Positive | 4 | 2 |
|  | Negative | 0 | 46 |

## Mitchell and George, 2020

Previously tested residual positive and negative nasopharyngeal patient samples collected in VTM and stored at −80°C were tested using the ID NOW EUA assay. Risk of recruitment bias is high because the specimens were selected by investigators, in part upon the basis of previous RT–PCR testing results. RT–PCR was performed using either a CDC or a New York State assay which had been granted Emergency Use authorization by the FDA. The risk of index test bias is high because dry swabs were not employed, and the risk of flow and timing bias is unclear (*Appendix 1—table 6*).

Investigators correlated the ID NOW performance with the Ct obtained during PCR. Although a 72% false-negative rate was found for patients whose Ct ranged from 35 to 40, no false-negatives were identified for patients with a Ct below 35.

**Appendix 1—table 6.** *Mitchell and George, 2020* Confusion matrix.

|  |  | ID NOW | |
| --- | --- | --- | --- |
|  |  | **Positive** | **Negative** |
| RT–PCR – either CDC or New York EUA | Positive | 33 | 13 |
|  | Negative | 0 | 15 |

## Moore et al., 2020

NP swabs in VTM were collected from a mix of ambulatory and hospitalized patients, some who were in the ICU. Some patient specimens were obtained consecutively, while, additional specimens were obtained by including samples in which SARS-CoV-2 RNA was detected by RT–PCR, and others were included in which virus had not been detected (*Appendix 1—table 7*). Risk of patient selection bias is high, while the risks of index test and flow and timing biases are unclear. Specimens were analyzed within 72 hr of collection and were held refrigerated at 4°C if all testing could not be completed on the same day.

In a separate evaluation, NP swabs were collected from 97 symptomatic emergency department patients who had negative results from a dry nasal swab tested at the point of care by ID NOW. These NP swabs were subsequently evaluated by R-TPCR on the Abbott m2000 system. SARS-CoV-2 RNA was detected in 13 of these NP swabs, or 13.4% of the total. The median Ct for these ID NOW – negative/m2000 positive cases was – 19.82.

**Appendix 1—table 7.** *Moore et al., 2020* Confusion matrix.

|  |  | ID NOW | |
| --- | --- | --- | --- |
|  |  | **Positive** | **Negative** |
| Abbott m2000 | Positive | 94 | 31 |
|  | Negative | 0 | 73 |

## Rhoads et al., 2020

A convenience sample of ninety-six clinical remnant specimens (11 supervised self-collected nasal swabs in normal saline and 85 provider-collected NP swabs in VTM) that had previously tested positive for SARS-CoV2 by RT–PCR were retested using ID Now, Diasorin Simplexa and a modified CDC LDT method (*Appendix 1—table 8*). The study was rated as having a high risk of patient selection bias, and unclear risk of both index test bias and flow and timing bias.

**Appendix 1—table 8.** *Rhoads et al., 2020* Confusion matrix.

|  |  | ID NOW | |
| --- | --- | --- | --- |
|  |  | **Positive** | **Negative** |
| Diasorin Simplexa | Positive | 90 | 6 |
|  | Negative | 0 | 0 |

## Smithgall et al., 2020

Investigators compared performance of Roche Cobas 6800, Cepheid Xpert and Abbott ID NOW assays using 88 residual NP swabs previously confirmed as positive that were chosen to represent the full range of observed Ct values, and 25 NP swabs previously confirmed as negative, all of which had been held at 4°C in VTM for no more than 48 hr prior to testing (*Appendix 1—table 9*). Risk of patient selection bias is rated as high due to the modified case-control design, and risk of index test bias and flow and timing bias are rated as unclear. The table below shows the confusion matrix for ID NOW compared to the Roche COBAS 6800, but the composite values shown in *Table 2* of the paper reflect the fact that the Cepheid Xpert identified two cases not identified by COBAS, while COBAS identified one case not identified by Xpert.

The authors demonstrated that agreement between ID NOW and RT–PCR was perfect for Cobas Ct ≤ 30, but fell off dramatically at higher Ct.

**Appendix 1—table 9.** *Smithgall et al., 2020* Confusion matrix.

|  |  | ID NOW | |
|---|---|---|---|
|  |  | **Positive** | **Negative** |
| Roche COBAS | Positive | 65 | 23 |
|  | Negative | 0 | 25 |

## Thwe and Ren, 2020

Investigators compared results from 182 paired dry NP swabs on ID NOW, and NP swabs in VTM, from symptomatic hospitalized and emergency department patients. Dry NP swabs were transported to the ID NOW testing area within 2 hr of collection, but total time to analysis was not reported. We rate the risk of patient recruitment bias to be high due to the inclusion of hospitalized inpatients (*Appendix 1—table 10*). We rate the risk of flow and timing bias, and index test bias, as unclear. PCR was carried out using any of several systems, including The real-time Abbott RealTime SARS-CoV-2 (Abbott Park, IL), Panther Fusion SARS-COV-2 (San Diego, CA), and Cepheid Xpert Xpress SARS-CoV-2 (Sunnyvale, CA) and a laboratory-developed test (LDT).

**Appendix 1—table 10.** *Thwe and Ren, 2020* Confusion matrix.

|  |  | ID NOW | |
|---|---|---|---|
|  |  | **Positive** | **Negative** |
| RT–PCR | Positive | 8 | 7 |
|  | Negative | 0 | 167 |

## Zhen et al., 2020b

This study included 108 NP swabs from symptomatic patients, of which 20 were collected prospectively and 88 were taken from a collection of frozen specimens (−80°C) that had been previously tested. All samples were tested using the Hologic Panther Fusion SARS-CoV-2 assay, the Abbott ID NOW assay and the Cepheid Xpert Xpress assay (*Appendix 1—table 11*). The prospective 20 specimens were processed fresh on each platform at the time of patient testing. Risk of patient selection bias was rated as unclear. Risk of index test bias was considered to be unclear, due to use of frozen specimens, and risk of flow and timing bias was also rated as unclear.

All false-negative ID NOW results were associated with Hologic Panther Ct ≥32.

Authors also performed a LOD analysis and found an LOD for ID NOW of 20,000 copies/mL, with LOD for Xpert Xpress of 100 copies/mL, and 1000 copies/mL for GenMark ePlex.

**Appendix 1—table 11.** *Zhen et al., 2020b* Confusion matrix.

| | | ID NOW | |
|---|---|---|---|
| | | Positive | Negative |
| Hologic Panther Fusion | Positive | 50 | 7 |
| | Negative | 0 | 50 |

## Comer and Fisk, 2020

This study sampled all COVID-19 symptomatic prospective hospital admissions with combined naso-pharyngeal (NP) and oropharyngeal (OP) swabs in the ED with the Abbott ID NOW and tested them immediately in the emergency department, if negative, recollect expeditiously and test on a Becton Dickinson BD MAX. Retesting occurred within a few hours (*Appendix 1—table 12*). The risk of patient selection bias appears to be high, based upon considering a group that is being considered for hospitalization, rather than the general population of symptomatic patients possibly suffering from SARS-CoV-2 infection. The risks of index test bias and flow and timing bias appear to be low.

**Appendix 1—table 12.** *Comer and Fisk, 2020* Confusion matrix.

| | | ID NOW | |
|---|---|---|---|
| | | Positive | Negative |
| Becton Dickinson BD MAX | Positive | 0 | 1 |
| | Negative | 0 | 116 |

## Ghofrani et al., 2020

Investigators employed a 'convenience sample' that included patients who had a RT–PCR sample collected close to the time of presentation followed by a re-swab for ID NOW, and those who were already known to be PCR-positive and whose residual NP samples were tested by ID NOW (*Appendix 1—table 13*). Some specimens employed in the ID NOW testing were dry, while others were transported in UTM. RT–PCR testing was conducted at one of several different laboratories, and the specific tests utilized were not reported. Risk of both patient selection bias and of flow and timing bias considered to be high. The risk of index test bias is rated as unclear.

**Appendix 1—table 13.** *Ghofrani et al., 2020* Confusion matrix.

| | | ID NOW | |
|---|---|---|---|
| | | Positive | Negative |
| RT–PCR | Positive | 16 | 1 |
| | Negative | 1 | 95 |

## SoRelle et al., 2020

This letter reports a study in which ID NOW as compared with Cepheid Xpert Xpress on 59 saliva samples from symptomatic subjects (*Appendix 1—table 14*). Details regarding collection environment and saliva transport are not provided. Investigators also compared saliva tested using the ID NOW system with NPS testing using RT–PCR (Abbott m2000); this data is not included in the present review. Results from a single test reported as 'invalid' on the ID NOW platform are not included in the confusion matrix below.

There is an uncertain risk of patient recruitment bias due to the lack of information. We have rated the risk of index test bias and flow and timing bias as low, based on the author's assertion that all testing was performed in accordance with the manufacturer's instructions.

Investigators noted that most ID NOW false negative results occurred in patients tested ≥2 weeks after symptom onset. They estimated the LOD for the ID NOW assay at 2000 copies/mL.

**Appendix 1—table 14.** *SoRelle et al., 2020* Confusion matrix.

| | | ID NOW | |
| --- | --- | --- | --- |
| | | **Positive** | **Negative** |
| Cepheid Xpert Xpress | Positive | 23 | 0 |
| | Negative | 0 | 35 |

# Appendix 2

## Statistical considerations for ID NOW COVID-19 study
### Overview

Total N = 2000 subjects at a single time point.
(Arm 1) N1 = 1500 symptomatic subjects.
(Arm 2) N2 = 500 asymptomatic subjects.

The reference method is a composite of two laboratory PCR methods (1) Hologic COVID-19 test (qualitative) and (2) CDC COVID-19 test (with CT values); reference positive is EITHER laboratory test positive; reference negative is BOTH laboratory tests negative.

(Arm 1) Primary goal is to estimate the sensitivity and specificity of the ID NOW COVID-19 test, as compared to the reference method defined above.

(Arm 2) Primary goal is to estimate the prevalence using ID NOW, adjusting for the sensitivity and specificity estimated in Arm 1.

## Power analysis for Arm 1

The study does not have formal acceptance criteria. However, for the purpose of powering the study, the following objectives are assumed.

- Lower limit of the two-sided 95% confidence interval > 92.00% for sensitivity.
- Lower limit of the two-sided 95% confidence interval > 95.00% for specificity.

The objective is to achieve 80% power assuming a sensitivity of 97.5% in the population.
Power Analysis of One Proportion
Numeric Results for testing H0: p=P0 versus H1: p>P0.
Test Statistic: Exact Test.

| | | Proportion proportion | | | | | |
| | | Given H0 | Given H1 | Target | Actual | | Reject H0 |
| Power | N | (P0) | (P1) | Alpha | Alpha | Beta | If R ≥ This |
| --- | --- | --- | --- | --- | --- | --- | --- |
| 0.8002 | 125 | 0.9200 | 0.9752 | 0.0250 | 0.0248 | 0.1998 | 121 |
| 0.8252 | 150 | 0.9200 | 0.9750 | 0.0250 | 0.0169 | 0.1748 | 145 |

N is the sample size drawn from the population. R/N is the point estimate of the proportion in the sample drawn from the population. The last column of the table is the minimum value of R required to achieve the objective of the study. Alpha is the probability of achieving the objective when the population proportion is P0. Power is the probability of achieving the objective when the population proportion is P1. Beta is 1 – Power.

A minimum of N = 125 reference positive subjects is recommended. Assuming 10% prevalence in Arm 1, the minimum recommended enrollment for Arm one is N = 1250. The actual enrollment into Arm one is expected to be N = 1500 subjects.

Assuming N = 1350 reference negative subjects, the specificity estimate in Arm one will meet the stated objective with high power, as shown in the calculation below.
Power Analysis of One Proportion
Numeric Results for testing H0: p=P0 versus H1: p>P0
Test Statistic: Exact Test

| | | Proportion proportion | | | | | |
| | | Given H0 | Given H1 | Target | Actual | | Reject H0 |
| Power | N | (P0) | (P1) | Alpha | Alpha | Beta | If R ≥ This |
| --- | --- | --- | --- | --- | --- | --- | --- |
| 0.9563 | 1350 | 0.9500 | 0.9700 | 0.0250 | 0.0196 | 0.0437 | 1299 |

## Power analysis for Arm 2

Estimating the prevalence of disease with an imperfect diagnostic test is a well-known statistics problem in the field of epidemiology (**Peter, 2011**; **Lewis and Torgerson, 2012**).

The relationship between the prevalence of the disease (θ) and the test positive rate of the diagnostic (φ) is given by the following equation

φ = (Se +Sp − 1) θ + (1 − Sp), Ref (**Peter, 2011**) Equation (2) where Se is the sensitivity and Sp is the specificity of the diagnostic test.

The relationship between θ and φ is linear with slope (Se + Sp − 1) and intercept (1 − Sp). The slope depends on both the sensitivity and specificity and varies between 0 (for a random test where Se + Sp = 1) and 1 (for a perfect test where Se = 1 and Sp = 1). The intercept is the depends only on the specificity (it is the probability of a false positive, conditional on the subject being disease negative).

For known values of Se and Sp, the 95% confidence interval of φ can be estimated from the binomial distribution and mapped to a corresponding 95% confidence interval in θ. However, if Se and Sp are estimated based on a clinical study, then the confidence interval in θ becomes wider due to the uncertainty in Se and Sp. Monte Carlo simulations can be used to calculate the confidence interval in θ based on the statistical estimation of the three binomial proportions, φ, Sp, and Se. Alternatively, **Basu et al., 2020** describes a Bayesian approach to the interval estimation of θ that accounts for the uncertainty in Sp and Se.

For this study, a simplified analytical method can be employed to calculate an approximate 95% confidence interval. The method is stated below.

θ = A/B
A = φ − (1 − Sp)
B = Se − (1 − Sp)

The 95% confidence interval for A is calculated by the Wilson score method for estimating the difference between two binomial proportions. The same method is applied to B. For the parameters of interest for this study, an approximate 95% confidence interval of theta is given by the 95% confidence interval of A divided by the point estimate of B.

Two-sided 95% confidence interval for the prevalence (θ) based on the test positive rate (φ) estimated in Arm two and the specificity (Sp) estimated in Arm 1 of the study*.

| | Arm2 | Arm1 | φ | (1 − Sp) | φ − (1 − Sp) | | | θ | |
| | | | P2 | P1 | P2 − P1 | | | | |
| Confidence level | N2 | N1 | Sample | Sample | Sample | Lower limit | Upper limit | Lower limit | Upper limit |
|---|---|---|---|---|---|---|---|---|---|
| 0.950 | 500 | 1500 | 0.050 | 0.050 | 0.000 | −0.0201 | 0.0248 | −0.0223 | 0.0276 |
| 0.950 | 500 | 1500 | 0.060 | 0.050 | 0.010 | −0.0115 | 0.0363 | −0.0128 | 0.0403 |
| 0.950 | 500 | 1500 | 0.070 | 0.050 | 0.020 | −0.0028 | 0.0476 | −0.0031 | 0.0529 |
| 0.950 | 500 | 1500 | 0.080 | 0.050 | 0.030 | 0.0060 | 0.0589 | 0.0067 | 0.0654 |
| 0.950 | 500 | 1500 | 0.090 | 0.050 | 0.040 | 0.0148 | 0.0700 | 0.0164 | 0.0778 |
| 0.950 | 500 | 1500 | 0.100 | 0.050 | 0.050 | 0.0237 | 0.0811 | 0.0263 | 0.0901 |
| 0.950 | 500 | 1500 | 0.020 | 0.020 | 0.000 | −0.0124 | 0.0175 | −0.0133 | 0.0188 |
| 0.950 | 500 | 1500 | 0.030 | 0.020 | 0.010 | −0.0044 | 0.0298 | −0.0047 | 0.0320 |
| 0.950 | 500 | 1500 | 0.040 | 0.020 | 0.020 | 0.0037 | 0.0418 | 0.0040 | 0.0449 |
| 0.950 | 500 | 1500 | 0.050 | 0.020 | 0.030 | 0.0120 | 0.0535 | 0.0129 | 0.0575 |
| 0.950 | 500 | 1500 | 0.060 | 0.020 | 0.040 | 0.0204 | 0.0651 | 0.0219 | 0.0700 |
| 0.950 | 500 | 1500 | 0.070 | 0.020 | 0.050 | 0.0290 | 0.0765 | 0.0312 | 0.0823 |

*This table assumes a fixed Se = 0.95. At a specificity of 95% in Arm one and a test positive rate of 8% in Arm 2, the two-sided 95% confidence interval of the asymptomatic prevalence is 0.67–6.54%. At a specificity of 98% in Arm one and a test positive rate of 5% in Arm 2, the two-sided 95% confidence interval of the asymptomatic prevalence is 1.29–5.75%. In conclusion, a prevalence of 3% can be detected with a sample size of N = 500 in the asymptomatic arm, provided that the sensitivity and specificity objectives are achieved in the symptomatic arm of the study.

