## [Decision Letter]

Our editorial process produces two outputs: i) public reviews designed to be posted alongside the preprint for the benefit of readers; ii) feedback on the manuscript for the authors, including requests for revisions, shown below.

Thank you for submitting your article "Evaluation of ID NOW and RT-PCR for Detection of SARS-CoV-2 in an Ambulatory Population" for consideration by *eLife*. Your article has been reviewed by 2 peer reviewers, and the evaluation has been overseen by a Reviewing Editor and Mone Zaidi as the Senior Editor. The following individual involved in review of your submission has agreed to reveal their identity: Ryan Phan (Reviewer #1).

While there was significant interest in the work there are several concerns that need to be addressed including the following comments

The cohort assessed in this study is small and localized. The data is undermined by sample size, with the most glaring example being the 100% negative percent agreement, which doesn't compare with the known performance of the test in broader populations.

Essential Revisions:

1. Address the sample size

2. The difference between the findings of this study and other published studies

Reviewer #1 (Recommendations for the authors):

There is no additional comment for the authors

Reviewer #2 (Recommendations for the authors):

Tu et al. submit a manuscript that evaluates the performance of the Abbott ID NOW SARS-CoV-2 test in an ambulatory cohort relative to RT-PCR tests. They enrolled 785 symptomatic patients, 21 tested positive for SARS-CoV-2 by ID NOW and PCR (Hologic) while 2 tested positive only via PCR. They also tested 189 asymptomatic individuals, none of whom tested positive by either ID NOW or PCR. The positive agreement between ID NOW and PCR was 91.3%, and the negative percent agreement was 100%. The authors also provide a review and meta-analysis of ID NOW performance across at least a dozen other named studies. While I found the review and meta-analysis of these other studies in comparison to the data the authors collected in their study thorough and interesting, I find the cohort assessed in this study to be far too small and localized to support publication of this study in this format. The data is too undermined by sample size to be a notable addition to the already crowded literature, not to mention in comparison to Abbott's public data. The most glaring example of this would be the 100% negative percent agreement finding in this study, which doesn't compare with the known performance of the test in broader application.

---

## [Author Response]

While there was significant interest in the work there are several concerns that need to be addressed including the following commentsThe cohort assessed in this study is small and localized. The data is undermined by sample size, with the most glaring example being the 100% negative percent agreement, which doesn't compare with the known performance of the test in broader populations.Essential Revisions:1. Address the sample size2. The difference between the findings of this study and other published studiesReviewer #2 (Recommendations for the authors):Tu et al. submit a manuscript that evaluates the performance of the Abbott ID NOW SARS-CoV-2 test in an ambulatory cohort relative to RT-PCR tests. They enrolled 785 symptomatic patients, 21 tested positive for SARS-CoV-2 by ID NOW and PCR (Hologic) while 2 tested positive only via PCR. They also tested 189 asymptomatic individuals, none of whom tested positive by either ID NOW or PCR. The positive agreement between ID NOW and PCR was 91.3%, and the negative percent agreement was 100%. The authors also provide a review and meta-analysis of ID NOW performance across at least a dozen other named studies. While I found the review and meta-analysis of these other studies in comparison to the data the authors collected in their study thorough and interesting, I find the cohort assessed in this study to be far too small and localized to support publication of this study in this format. The data is too undermined by sample size to be a notable addition to the already crowded literature, not to mention in comparison to Abbott's public data. The most glaring example of this would be the 100% negative percent agreement finding in this study, which doesn't compare with the known performance of the test in broader application.

We greatly thank the reviewer for noting the differences between the current study and those previously published regarding negative percent agreement. We apologize that the manuscript was not written clearly enough to delineate that the current cohort study is larger than any other single study reported in the literature on ID NOW and is the only study in which power analysis was used to plan sample size. We have revised the manuscript to emphasize that the current study is the largest of its kind to analyze the ID NOW SARS-CoV-2 PCR test and believe that this remains an important distinction from other papers in the literature. Importantly, we have not submitted the current cohort study as a stand-alone clinical study, but rather as a clinical study associated with a systematic review and meta-analysis; for this, we have included as much publicly available and publicly described data as possible to maximize the utility to the practitioner. From this open analysis, only three studies exist with a low risk of patient selection bias (including the current one).

We thank the reviewer for suggesting a review of additional studies available on Abbott’s website (https://abbott.mediaroom.com/2020-05-21-Abbott-Releases-Interim-Clinical-Study-Data-on-ID-NOW-COVID-19-Rapid-Test-Showing-Strong-Agreement-to-Lab-Based-Molecular-PCR-Tests). We have re-examined the details of the studies reported by Abbott as well as those used by FDA and note that all of them have an overall sample size less than that used in the current clinical study. Some of the studies cited in Abbott’s link are similar to the current findings; for example, one “urgent care study” found one case where ID NOW PCR was positive, and RT-PCR was negative from a group of 256 patients overall giving a rate of >99% negative agreement. Abbott’s site also reports on an unpublished study in which ID NOW demonstrated 94.7% positive agreement and 98.6% negative agreement compared to the Centers for Disease Control (CDC) 2019-Novel Coronavirus (COVID-19) Real-Time RT-PCR Diagnostic Panel, as well as on a study of hospitalized and nursing home patients where ID NOW showed 85.7% positive agreement and 97.6% negative agreement compared to the CDC assay, and 83.3% positive agreement and 96.5% negative agreement when compared to Hologic's Panther Fusion SARS-CoV-2 PCR test. We agree with the reviewer that the negative percent agreements in these studies does appear to differ from the current study; however, those results also differ from the summary results of the published studies that we included in the systematic review that showed an overall 99.8% negative agreement.

As detailed above, the negative percent agreement is one area of potential confusion for readers and we are very grateful to the reviewer for helping to point this out. Many of these studies and press releases are unpublished or utilize interim clinical study data, thereby making it difficult to be certain of the exact reasons for differences in positive and negative percent agreement. What is evident, however, is that the effectiveness of each of these technologies in detecting SARS-CoV-2 virus is highly dependent on sample acquisition site and timing of testing, particularly with ID NOW, as well as specimen transport and to a lesser extent swab construction. To help clarify differences that exist in the literature we have expanded the Discussion to clarify and explain these issues. The relevant section of the revised Discussion now reads as follows:

Although the performance of ID NOW in an asymptomatic population has not been established, and caution may be appropriate when using ID NOW with a high-risk population, increased frequency of testing, together with a rapid turnaround time, are likely to have greater impact on population health outcomes than are differences in test sensitivity (Larremore et al., 2020; Mina et al., 2020).